# Robotic Major Hepatectomy in Elderly Patient

**DOI:** 10.3390/cancers16112083

**Published:** 2024-05-30

**Authors:** Antonella Delvecchio, Maria Conticchio, Riccardo Inchingolo, Francesca Ratti, Paolo Magistri, Andrea Belli, Graziano Ceccarelli, Francesco Izzo, Marcello Giuseppe Spampinato, Nicola De’ Angelis, Patrick Pessaux, Tullio Piardi, Fabrizio Di Benedetto, Luca Aldrighetti, Riccardo Memeo

**Affiliations:** 1Unit of Hepato-Biliary and Pancreatic Surgery, “F. Miulli” General Hospital, Acquaviva delle Fonti, 70021 Bari, Italy; antodel88@libero.it (A.D.); m.conticchio@miulli.it (M.C.); r.memeo@miulli.it (R.M.); 2Unit of Interventional Radiology, “F. Miulli” General Hospital, Acquaviva delle Fonti, 70021 Bari, Italy; 3Hepatobiliary Surgery Division, IRCCS San Raffaele Scientific Institute, 20132 Milano, Italy; ratti.francesca@hsr.it (F.R.); aldrighetti.luca@hsr.it (L.A.); 4Hepatobiliary Surgery Division, Vita-Salute San Raffaele University, 20132 Milano, Italy; 5Unit of Hepato-Pancreato-Biliary Surgery and Liver Transplantation, University of Modena and Reggio Emilia, 41121 Modena, Italy; paolo.magistri@unimore.it (P.M.); fabrizio.dibenedetto@unimore.it (F.D.B.); 6Unit of Hepato-Biliary and Pancreatic Surgery, Istituto Nazionale Tumori IRCCS Fondazione G. Pascale, 80131 Napoli, Italy; a.belli@istitutotumori.na.it (A.B.); f.izzo@istitutotumori.na.it (F.I.); 7Unit of General Surgery, San Giovanni Battista Hospital, USL Umbria 2, 06034 Foligno, Italy; graziano.ceccarelli@uslumbria2.it; 8Unit of General Surgery, “Vito Fazzi” Hospital, 73100 Lecce, Italy; marcello.spampinato@gmail.com; 9Unit of Digestive and Hepatobiliary Surgery, Centre Hospitalier Universitaire Henri Mondor, 94000 Créteil, France; nicola.deangelis@unife.it; 10Department of Visceral and Digestive Surgery, Unit of Hepato-Bilio-Pancreatic Surgery, Nouvel Hospital Civil, University Hospital of Strasbourg, 67000 Strasbourg, France; patrick.pessaux@chru-strasbourg.fr; 11Unit of Surgery, Hôpital Robert Debré, 51100 Reims, France; tpiardi@chu-reims.fr

**Keywords:** robotic liver surgery, major hepatectomy, elderly patient

## Abstract

**Simple Summary:**

Robotic liver surgery is becoming the future of minimally invasive surgery, overcoming the intrinsic limitations of laparoscopic surgery and allowing the execution of highly complex procedures such as major hepatectomies. Despite these advantages, up to now, no evidence of benefits associated with the robotic approach in liver surgery has been clearly defined in the literature. Data from a multicentric retrospective database including 131 major liver resections in nine European hospital centers were analyzed. The patients were divided into two groups based on age: elderly and young. Perioperative data were compared between the two groups. The aim of this study was to analyze the feasibility and safety of robotic major liver resection in elderly patients.

**Abstract:**

Background: the role of minimally invasive liver surgery has been progressively developed, with the practice increasing in safety and feasibility also with respect to major liver resections. The aim of this study was to analyze the feasibility and safety of major liver resection in elderly patients. Methods: data from a multicentric retrospective database including 1070 consecutive robotic liver resections in nine European hospital centers were analyzed. Among these, 131 were major liver resections. Patients were also divided in two groups (<65 years old and ≥65 years old) and perioperative data were compared between the two groups. Results: a total of 131 patients were included in the study. Operative time was 332 ± 125 min. Postoperative overall complications occurred in 27.1% of patients. Severe complications (Clavien Dindo ≥ 3) were 9.9%. Hospital stay was 6.6 ± 5.3 days. Patients were divided into two groups based on their age: 75 patients < 65 years old and 56 patients ≥ 65 years old. Prolonged pain, lung infection, intensive care stay, and 90-day readmission were worse in the elderly group. The two groups were matched for ASA and Charlson comorbidity score and, after statistical adjustment, postoperative data were similar between two groups. Conclusions: robotic major liver resection in elderly patients was associated with satisfying short-term outcomes.

## 1. Introduction

Minimally invasive liver surgery (MILS) has progressively developed, increasing in safety and feasibility even with respect to major liver resections. Robotic techniques are becoming the future of minimally invasive surgery, overcoming the intrinsic limitations of laparoscopic surgery. The advantages of robotic surgery concern the possibility of both easily performing complex procedures and improving surgical techniques.

Complex procedures that can be facilitated thanks to the robotic approach include major hepatectomies, resections of difficult segments, and biliary anastomosis [1,2,3]. The improvement of surgical techniques concern tremor filter, endowristed movements, augmented surgeon’s ergonomy, 3D high-definition vision, and faster learning curves [4,5]. Despite these advantages, the absence of tactile feedback, high costs, limited access to the robotic platform, and few surgical instruments available limit robotic development and application.

The benefits of robotic liver resection are emerging thanks to studies that confirm the role of MILS in reducing postoperative complications, providing better postoperative recovery, and shortening hospitalization stays [6,7]. These represent a real advantage, especially in elderly patients who can benefit from a quick return home [8]. To date, the literature is poor in studies that investigate the role of robotic major hepatectomy in elderly patients. 

The aim of this study is to analyze the feasibility and safety of robotic major hepatectomy (RMH) in elderly patients, considering intraoperative and postoperative short term outcomes.

## 2. Materials and Methods

Data were collected from a multicentric retrospective database including 1070 consecutive robotic liver resections for benign and malignant liver tumors. Patients were enrolled in nine European hospital centers from 2011 to 2022. Each center had ethics committee approval and informed consent to collect patient clinical data. From the entire series, 131 RMH were performed. Patients who underwent RMH were divided into two groups based on their age: young < 65 and elderly ≥ 65 years. Patients over 65 years old were considered elderly. Patients < 18 years old, ASA > IV, and examples of minor liver resections were excluded from the study. Preoperative, intraoperative, and postoperative data were retrospectively analyzed and compared in both groups.

### 2.1. Preoperative Data

Preoperative data, in terms of patient characteristics and tumor characteristics, were recorded and analyzed for each included patient: age, gender, body mass index (BMI), previous abdominal surgery, comorbidity, liver fibrosis, Charlson comorbidity score, preoperative chemotherapy, indication to surgery, size and position of liver tumor. Preoperative chemotherapy was based on tumor biology. All patients were staged preoperatively with CT scans of chest–abdomen–pelvis and abdominal RMI when necessary. CT scans allowed to define the tumor diameter and vascular contact. A liver biopsy was performed in case of an unclear radiological diagnosis in accordance with the European Association for the Study of the Liver (EASL) [9]. The preoperative evaluation also included the surgical risk calculation considering the scoring system from the American Society of Anesthesiologists (ASA) and a liver assessment with the model for end-stage liver disease (MELD) scores. All the clinical cases were discussed in a multidisciplinary team.

### 2.2. Intraoperative Data 

The surgical procedure was performed by expert surgeons with a consolidated experience in minimally invasive and open hepatobiliary surgery. Liver segmentation anatomy and liver resection type were defined using the Couinaud classification and the Brisbane 2000 terminology, respectively [10,11]. Major liver resection was considered to be a resection of three or more contiguous Couinaud segments [12]. We included robotic left hepatectomy (RLH), robotic right hepatectomy (RRH), robotic extended left hepatectomy (RELH), robotic extended right hepatectomy (RERH), and robotic central hepatectomy (RCH). An intraoperative ultrasound (IOUS) was systematically performed at the beginning of the procedure. IOUS is a recommended tool to investigate liver anatomy to confirm number, size, and location of the tumor and to plan transection lines and margins. 

A parenchymal liver transection was performed with different types of devices based on the individual surgeon’s preference such as electrocoagulation, ultrasonic dissector, radiofrequency, or combined energy devices. The Pringle maneuver was routinely prepared at the beginning of the procedure and used according to the experience of each center. Data regarding operative time, estimated blood loss, blood transfusion, conversion rate to laparoscopic or open surgery, and pedicle clamping were recorded. Estimated blood loss was measured as the difference between the fluid present in the suction receptor and the lavage solution infused into the abdominal cavity.

### 2.3. Postoperative Data and Short-Term Outcomes

Postoperative data were recorded, including postoperative complications according to the Clavien Dindo grading system [13], reintervention, intensive care unit (ICU) stay, hospital stay, R1 resection, and readmission after 90 days. Prolonged pain was defined as the continued use of analgesics beyond the third postoperative day.

### 2.4. Statistical Analysis

Data were reported as mean and standard deviation or as frequency with percentage. Associations between categorical variables were evaluated using a chi-squared test or Fisher’s exact test as appropriate. Continuous variables were compared using the student’s *t*-test. Multivariate logistic and linear regressions were used to evaluate the independent association between the age category and dichotomous (logistic) or continuous (linear) variables, adjusting for ASA and the Charlson comorbidity score that were significantly different among patients younger and older than 65 years. A *p* value < 0.05 was considered to be statistically significant. All analyses were conducted using the STATA software, version 16 (Stata-Corp LP, College Station, TX, USA).

## 3. Results

### 3.1. Patients’ Characteristics and Perioperative Data

During the study period, we enrolled a total of 1070 consecutive robotic liver resections, 131 of which were robotic major hepatectomies. Patients’ characteristics and perioperative data of RMH are shown in Table 1. The mean age was 64 years and 59.5% were male patients; mean BMI was 26.6 kg/m^2^. A total of 28.2% of patients were ASA III. The Charlson comorbidity score mean was 4.9. Thirty-four patients had a previous abdominal surgery and were divided into previous open surgery (14, 10.7%) and previous laparoscopic surgery (20, 15.3%). No patients had more than two comorbidities. A total of 28.3% of patient had liver fibrosis with a mean MELD score of 6.8. The main indications for surgery were CRLM (colorectal liver metastasis) and HCC (hepatocellular carcinoma), with 39 (30%) and 37 (28.2%) patients, respectively (Figure 1). The mean size of the largest lesions was 47 ± 30 mm. A total of 30.2% of the lesions were distant from the vascular structures, while 20.9% of the lesions were in contact with the portal branch and 30.2% were in contact with the hepatic vein. Among the major hepatectomies, 70 (53.4%) were left hepatectomies, 57 (43.5%) were right hepatectomies, three (2.3%) were central hepatectomies, one (0.8%) was an extended left hepatectomy, and no extended right hepatectomies were recorded (Figure 2). All conversions were to the open approach, with a total number of 14 (10.7%). Operative time was 332 ± 125 min. Five patients (4.2%) had an intraoperative blood transfusion. In 68.7% of cases, a pedicle clamping was performed. The overall postoperative complication rate was 27.1%, and the most represented complications were pulmonary infection, prolonged pain, and biliary leakage: 9.9%, 5.3%, and 5.3%, respectively. Severe complications occurred in 9.9% of patients. No patient was reoperated. Hospital stay duration was 6.6 ± 5.3 days. Histological specimens had a mean free margin of 8 ± 8 mm. A total of 7.2% RMH were R1 parenchymal. Seven patients (5.3%) had complications after discharge and seven patients (5.3%) were readmitted to the hospital within 90 days after discharge.

### 3.2. Comparison of Young Patients versus Elderly Patients

Patients were divided into two groups based on their age: 75 patients were aged < 65 years and 56 patients were aged ≥ 65 years old. In terms of preoperative characteristics, a greater number of elderly patients were ASA III compared to the young group (42.9% vs. 17.3%, *p* 0.001) and had a higher Charlson comorbidity score (5.8 vs. 4.3, *p* 0.001). Cardiovascular diseases were more common in the elderly group compared to the young group (33.9% vs. 14.7%, *p* 0.009) (Table 2). There were no differences in terms of intraoperative data between the two groups (Table 3). The postoperative data highlighted a higher incidence of prolonged pain in the elderly group compared to the young group (12.5 vs. 0%, *p* 0.002) and a major number of pulmonary infections in the elderly group (16.1 vs. 5.3%, *p* 0.042). In the elderly group, a longer ICU stay was reported compared to the young group (0.79 ± 1.17 vs. 0.44 ± 0.70 days, *p* 0.042). Readmission within 90 days was more frequent in the elderly group compared to the young group (10 vs. 1.3%, *p* 0.042). After matching the two groups for ASA and Charlson comorbidity score thanks to statistical adjustment, there were no more differences between the two groups in terms of prolonged pain, pulmonary infection, ICU stay, and readmission at 90 days (*p* 0.509, 0.883, 0.256, and 0.460, respectively) (Table 4).

## 4. Discussion

The development of minimally invasive liver surgery has progressively improved, with the approach increasing in safety and feasibility also with respect to major liver resections. Use of robotic platforms has increased exponentially in the last 10 years, and many studies have been published on the subject. The robotic technique is becoming the future of minimally invasive surgery, overcoming the intrinsic limitations of laparoscopic surgery and allowing highly complex procedures to be performed [3]. In 2018, the first International Consensus Statement on robotic liver surgery was published [6]. However, the lack of high-quality evidence on the benefits of the robotic liver resection had not allowed for widespread implementation. In 2023, the new international experts consensus guidelines on robotic liver resection [7] aimed to update clinical recommendations to increase the implementation rate of robotic liver resections (RLRs). According to the latest consensus, the robotic approach for major hepatectomies is as safe and feasible as its laparoscopic and open counterparts. RLR has been shown to perform significantly better in terms of estimated blood loss compared to laparoscopic liver resection (LLR) and open liver resection (OLR). RLR has a lower conversion rate compared to LLR and a shorter hospital stay compared to OLR. The weak point of RLR compared to the other two techniques is the operating time, which was found to be longer in our study. The main advantages of robotic surgery concern the improvement of surgical techniques: tremor filter, dexterity given by the endowristed movements, augmented surgeon’s ergonomy, 3D high-definition vision, use of indocyanine green images, intraoperative liver navigation, and faster learning curves [4,5]. These benefits allow complex procedures such as major hepatectomies or resections in the posterosuperior segments to be performed [1,2]. On the other hand, the dissemination and application of robotic platforms have been limited by the absence of tactile feedback, their high costs, and the scarcity of available surgical instruments. The main aim of minimally invasive surgery is to improve patient outcomes, and this is achieved through the reduction of postoperative complications, performance of small skin incisions, a faster postoperative recovery, and a rapid return home. The benefits of these techniques are greater for elderly and frail patients [8]. The definition of elderly people concerns patients over 65 years old. In some studies, elderly patients were divided into groups as follows: “young old” patients over 65 but under 75 years of age, “intermediate old” patients over 75 but under 85 years of age, and “oldest old” patients over 85 years of age [14,15]. The increase in life expectancy and the accumulation of neoplastic pathologies over the years has led to an increase in the demand for surgeries on elderly patients. For this reason, the elderlies are an object of interest in the literature [16].

Our study shows the feasibility and safety of robotic major liver resections. These benefits were confirmed also for elderly patients after a comparative analysis with the young population.

A total of 131 RMH were analyzed. We performed a further analysis to compare the outcomes between young and elderly populations, with a cut off age of 65. Considering the total population, a mean Charlson comorbidity score of 4.9 was found. A score above 5 is generally an expression of significant clinical commitment. In our total series, 10.7% of patients had previous open abdominal surgery and 15.3% had previous laparoscopic abdominal surgery. Previous abdominal surgery causes abdominal adhesions and is associated with technical difficulties, but it does not represent a risk factor for longer operating times or increased postoperative complications and can be performed safely with a robotic approach [17]. Many patients, 28.3%, had underlying liver fibrosis with a mean MELD score of 6.8 and a satisfactory liver function. Some authors have demonstrated that RLR can be used for major liver resections in patients affected by cirrhotic disease, obtaining less postoperative pain and shorter hospital stays without changing oncological outcomes [16,18]. Despite the difference not being statistically significant, a lower incidence of patients with a benign pathology was found in the elderly group. The presence of a higher proportion of patients with malignant tumors plays a role in determining the risk of intra- and postoperative events compared to younger patients where benign indications generally occur in very healthy patients [19]. The main indications for RMH were colorectal liver metastasis, CRLM (30%), and HCC (28.2%). The rate of conversion was 0% to laparoscopy and 10.7% to the open approach in the total population, while it was 0% to laparoscopy and 12.5% to the open approach in the elderly population. Thanks to the high degree of movement and stereoscopic camera, the robotic platform provides advantages with respect to the management of intraoperative complications, such as bleeding or difficult exposure, often reported as causes of conversion in laparoscopic procedures [20]. According to the international experts consensus guidelines on robotic liver resection published in 2023 [7], the robotic approach for major liver resections is significantly better in terms of conversion rate compared to laparoscopic surgery. Yoshino et al. [21] showed that, for major hepatectomies in elderly patients, the laparoscopic approach had a conversion rate to open of 30%, compared to 2.6% for the robotic approach. An important benchmark in liver surgery concerns blood loss. Most series in the literature demonstrate a reduction in blood loss with the robotic approach compared to the laparoscopic one and this was more marked in major hepatectomies [7,22,23]. The robotic platform, thanks to endowrist technology, freedom of movement, and three-dimensional optics, can avoid injuries to major vessels. It allows for accurate identification and dissection of vessels for inflow and outflow control during liver resection [24]. In our series, the operative time was 332 min in the total population and 358 min in the elderly population, higher than that reported in other studies analyzing major hepatectomies in elderly patients. Yoshino [19] showed an average of 249 min and Sucandy [23] an average of 230 min in the elderly population. Chong [25] reported an average of 317 min for robotic right and extended right hepatectomy considering the entire population without age differences. In our total series, overall postoperative complications occurred in 27.1% of patients, with a severe complication rate of 9.9%. Chong [25] showed similar results when comparing robotic and laparoscopic right and extended right hepatectomies, with a postoperative morbidity and major postoperative morbidity of 30.9% and 10.9%, respectively, for the robotic approach. Instead, considering our two groups, the overall postoperative complication rate in the elderly group was 32.1%, with a severe postoperative complication rate of 14.3%, both of which were not statistically different from the rates recorded in the young population. In the literature, the reported morbidity rate of major hepatectomy in the elderly population is greater than 40% [21]. A minimally invasive approach can reduce the rate of postoperative complications, especially in frail patients [8,26]. Indeed, Zhang proved that robotic liver resection in elderly patients was associated with a lower complication rate compared with the open liver resection (7.0% vs. 17.4% after PSM, *p* = 0.015) [16]. Our results show a higher rate of postoperative pulmonary infection in the elderly group. Several studies have reported that elderly patients have a higher incidence of cardiopulmonary diseases [16,27,28]. With increasing age, the lung structure changes due to the loss of collagen and elasticity of the parenchyma, predisposing individuals to this complication. The average hospital stay was 6.6 days in our total series and 6 days in the elderly patients, 4 days longer than that reported by Yoshino for elderly patients [21]. Sucandy [23] reported an average hospital stay of 5 days for elderly patients and 4 days for young patients. ICU stay and 90 days readmission rates were higher in the elderly group, something which is explained by the fact that they are fragile patients with numerous comorbidities, a lower functional reserve, and a high rate of postoperative morbidity and mortality [29].

We homogenized the two groups (elderly and young patients) for ASA and Charlson comorbidity score and, after adjustment, the variables prolonged pain, lung infection, intensive care stay, complications after hospital discharge, and readmission at 90 days were no longer significantly different between the two groups. It follows that, regardless of age, comorbidities influence postoperative outcomes. This work has some limitations, first of all regarding the retrospective enrollment and nonrandomized nature. Secondly, the absence of long-term oncological outcomes due to ‘en cours’ follow-up. Thirdly, the enrollment of all liver diseases both benign and malignant. Fourthly, the learning curve and changes in indication for RLR could not be evaluated during the long study period for each center.

## 5. Conclusions

Robotic major liver resection can be considered a valid technique even in elderly patients, as it is associated with satisfying short-term outcomes.

## Figures and Tables

**Figure 1 cancers-16-02083-f001:**
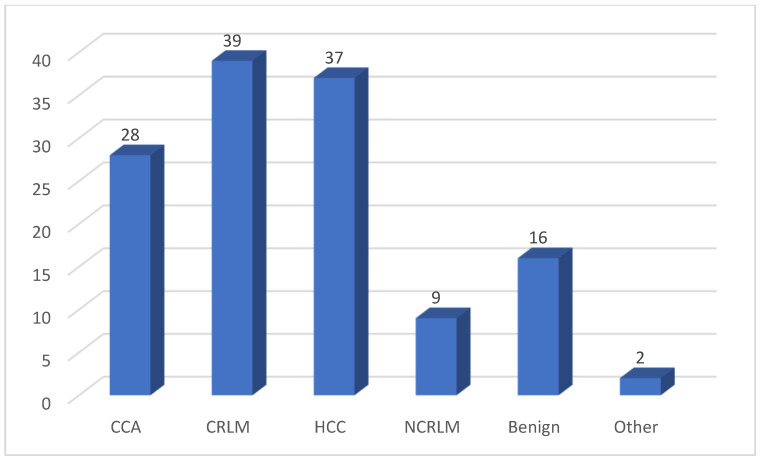
Indication to surgery. CCA: cholangiocarcinoma; CRLM: colorectal liver metastasis; HCC: hepatocellular carcinoma; NCRLM: non-colorectal liver metastasis.

**Figure 2 cancers-16-02083-f002:**
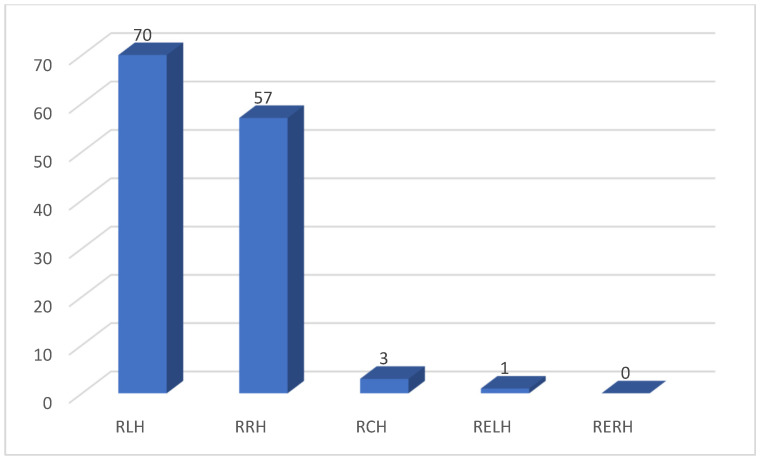
Type of liver resection. RLH: robotic left hepatectomy; RRH: robotic right hepatectomy; RCH: robotic central hepatectomy; RELH: robotic extended left hepatectomy; RERH: robotic extended right hepatectomy.

**Table 1 cancers-16-02083-t001:** Patients’ characteristics and perioperative data of robotic major hepatectomies (RMHs).

Variables	RMH
	n = 131
Age (yr), mean ± SD	64 ± 13
Male, n (%)	78 (59.5%)
BMI (kg/m^2^), mean ± SD	26.6 ± 3.6
ASA III score, n (%)	37 (28.2%)
ASA IV score, n (%)	0 (0.0%)
Charlson comorbidity score, mean ± SD	4.9 ± 2.4
Previous open abdominal surgery, n (%)	14 (10.7%)
Previous laparoscopic abdominal surgery, n (%)	20 (15.3%)
Comorbidity > 2, n (%)	0 (0.0%)
Liver fibrosis, n (%)	36 (28.3%)
MELD Score, mean ± SD	6.8 ± 1.6
Preoperative chemotherapy, n (%)	38 (29.0%)
Number of lesions, mean ± SD	1.79 ± 1.43
Tumor: size of biggest lesion (mm), mean ± SD	47 ± 30
Lesion non in contact with vessels, n (%)	39 (30.2%)
Lesion in contact with vessels: portal branch, n (%)	27 (20.9%)
Lesion in contact with vessels: hepatic vein, n (%)	39 (30.2%)
Conversion to open, n (%)	14 (10.7%)
Conversion to laparoscopy, n (%)	0 (0.0%)
Operative time (min), mean ± SD	332 ± 125
Estimated blood loss (mL), mean ± SD	282 ± 215
Blood transfusion, n (%)	5 (4.2%)
Pedicule clamping, n (%)	90 (68.7%)
Post operative complication, n (%)	35 (27.1%)
Biliary leakage, n (%)	7 (5.3%)
Hemorrhage, n (%)	2 (1.5%)
Prolonged pain, n (%)	7 (5.3%)
Ascitis, n (%)	5 (3.8%)
Pulmonary infection, n (%)	13 (9.9%)
Severe complication (Clavien ≥ 3), n (%)	13 (9.9%)
Re-intervention, n (%)	0 (0.0%)
ICU stay (days), mean ± SD	0.60 ± 0.94
Total hospital stay (days), mean ± SD	6.6 ± 5.3
Surgical margins (mm), mean ± SD	8 ± 8
R1 parenchymal, n (%)	7 (7.2%)
R1vasc, n (%)	0 (0.0%)
Complications after hospital discharge, n (%)	7 (5.3%)
Readmission at 90 days, n (%)	7 (5.3%)

SD: standard deviation; BMI: body mass index; ASA: American Society of Anesthesiologists; MELD: mayo end stage liver disease; ICU: intensive care unit.

**Table 2 cancers-16-02083-t002:** Patients and tumor characteristics in young patients compared to elderly patients.

	Age (Years)		
	<65	≥65	
	n = 75	n = 56	*p*
Age (yr), mean ± SD	54 ± 10	74 ± 5	<0.001
Male, n (%)	45 (60.0%)	33 (58.9%)	0.902
BMI (kg/m^2^), mean ± SD	26.4 ± 3.7	26.9 ± 3.5	0.479
ASA III score, n (%)	13 (17.3%)	24 (42.9%)	0.001
ASA IV score, n (%)	0 (0.0%)	0 (0.0%)	
Charlson comorbidity score, mean ± SD	4.3 ± 2.4	5.8 ± 2.2	0.001
Previous open abdominal surgery, n (%)	6 (8.0%)	8 (14.3%)	0.249
Previous laparoscopic abdominal surgery, n (%)	6 (8.0%)	14 (25.0%)	0.007
Cardiovascular disease, n (%)	11 (14.7%)	19 (33.9%)	0.009
Pulmonary diseases, n (%)	5 (3.8%)	3 (5.4%)	0.651
Liver fibrosis, n (%)	19 (26.4%)	17 (30.9%)	0.575
MELD Score, mean ± SD	6.4 ± 0.7	7.1 ± 2.0	0.113
Preoperative chemotherapy, n (%)	21 (28.0%)	17 (30.4%)	0.769
Indication to surgery			
CCA, n (%)	16 (21.3%)	12 (21.8%)	0.947
CRLM, n (%)	23 (30.7%)	16 (29.1%)	0.846
HCC, n (%)	16 (21.3%)	21 (38.2%)	0.035
NCRLM, n (%)	8 (10.7%)	1 (1.8%)	0.078
Benign, n (%)	12 (16.0%)	4 (7.3%)	0.135
Other, n (%)	0 (0.0%)	1 (1.8%)	0.423
Number of lesions, mean ± SD	1.8 ± 1.4	1.8 ± 1.5	0.899
Tumor: size of biggest lesion (mm), mean ± SD	50 ± 32	42 ± 26	0.161
Lesion non in contact with vessels, n (%)	19 (25.3%)	20 (37.0%)	0.153
Lesion in contact with vessels: portal branch, n (%)	16 (21.3%)	11 (20.4%)	0.894
Lesion in contact with vessels: hepatic vein, n (%)	26 (34.7%)	13 (24.1%)	0.196

SD: standard deviation; BMI: body mass index; ASA: American Society of Anesthesiologists; MELD: mayo end stage liver disease; CCA: cholangiocarcinoma; CRLM: colorectal liver metastasis; HCC: hepatocellular carcinoma; NCRLM: non colorectal liver metastasis.

**Table 3 cancers-16-02083-t003:** Intraoperative data relative to young patients compared to elderly patients.

	Age (Years)		
	< 65	≥ 65	
	n = 75	n = 56	*p*
Type of liver resection			
Left hepatectomy, n (%)	39 (52.0%)	31 (55.4%)	0.703
Right hepatectomy, n (%)	35 (46.7%)	22 (39.3%)	0.399
Central hepatectomy, n (%)	1 (1.3%)	2 (3.6%)	0.576
Extended left hepatectomy, n (%)	0 (0.0%)	1 (1.8%)	0.427
Extended right hepatectomy, n (%)	0 (0.0%)	0 (0%)	
Conversion to open, n (%)	7 (9.3%)	7 (12.5%)	0.562
Conversion to laparoscopy, n (%)	0 (0.0%)	0 (0.0%)	
Operative time (min), mean ± SD	312 ± 109	358 ± 140	0.050
Estimated blood loss (mL), mean ± SD	282 ± 196	283 ± 241	0.968
Blood transfusion, n (%)	3 (4.4%)	2 (3.9%)	1.000
Pedicule clamping, n (%)	54 (72.0%)	36 (64.3%)	0.346

**Table 4 cancers-16-02083-t004:** Postoperative data relative to young patients compared to elderly patients.

	Age (Years)			
	< 65	≥ 65		Adjusted
	n = 75	n = 56	*p*	*p*
Postoperative complications, n (%)	17 (23.3%)	18 (32.1%)	0.262	
Biliary leakage, n (%)	3 (4.0%)	4 (7.1%)	0.460	
Hemorrhage, n (%)	1 (1.3%)	1 (1.8%)	1.000	
Prolonged pain, n (%)	0 (0.0%)	7 (12.5%)	0.002	0.509
Ascitis, n (%)	4 (5.3%)	1 (1.8%)	0.392	
Pulmonary infection, n (%)	4 (5.3%)	9 (16.1%)	0.042	0.883
Severe complication (Clavien ≥ 3), n (%)	5 (6.7%)	8 (14.3%)	0.149	
Re-intervention, n (%)	0 (0.0%)	0 (0.0%)		
ICU stay (days), mean ± SD	0.44 ± 0.70	0.79 ± 1.17	0.042	0.256
Total hospital stay (days), mean ± SD	7 ± 6	6 ± 4	0.487	
Surgical margins (mm), mean ± SD	7 ± 8	9 ± 9	0.430	
R1 parenchymal, n (%)	4 (7%)	3 (8%)	1.000	
R1vasc, n (%)	0 (0%)	0 (0%)		
Complications after hospital discharge, n (%)	1 (1.3%)	6 (10.7%)	0.042	0.046
Readmission at 90 days, n (%)	1 (1.3%)	6 (10.7%)	0.042	0.460

ICU: intensive care unit; SD: standard deviation.

## Data Availability

The data presented in this study are available on request from the corresponding author due to privacy reasons.

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
