# Peer review of "Robotic Major Hepatectomy in Elderly Patient"

_cancers, 2024, doi:10.3390/cancers16112083_

Round 1

Reviewer 1 Report (Previous Reviewer 2)

Comments and Suggestions for Authors

The aimed of the study was to analyze the feasibility and safety of robotic major liver resection in elderly patients.

We congrats the authors. The majority of the issues have been addressed according to the reviewers' suggestions; however, one concern need to be addressed before consideration for publication.

Major comments

One of the major concerns was that although the mean age of the entire cohort was 64 years, the cutoff for elderly patients was considered to be 65. The manuscript was revised, and the authors added that the definition of elderly people in the present study concerned people over 65 years of age. With this in mind, could you please elaborate on why you considered ≥65 years as the criterion to define elderly patients? Or maybe add a reference to this definition.

Author Response

Dear Reviewer,

Thank you for the consideration of this paper and for your comment.

We modified the following sentences and we add the following references:

The definition of elderly people concerns patients over 65 years old. In some studies, elderly patients were divided as follow: “young old” patients over 65 but under 75 years of age, “intermediate old” patients over 75 but under 85 years of age, and “oldest old” patients over 85 years of age.

References

Campion EW. (1994) The oldest old. N Engl J Med. https://doi.org/10.1056/NEJM199406233302509.

Orimo H. (2006) Reviewing the definition of elderly. Jpn J Geriatr. https://doi.org/10.3143/geriatrics.43.27.

Reviewer 2 Report (New Reviewer)

Comments and Suggestions for Authors

The authors have addressed the criticisms raised by the reviewer and the manuscript has been improved.

Author Response

Dear Reviewr,

Thank you for the consideration of this paper and for your comment.

regards

This manuscript is a resubmission of an earlier submission. The following is a list of the peer review reports and author responses from that submission.

Round 1

Reviewer 1 Report

Comments and Suggestions for Authors

Robotic major hepatectomy is a very relevant topic, given current results promise significant advantages for the patients.

The introduction is held pleasantly short and targets the topic. The authors should be more precise in the wording of the target of their study (line #62/63), describing more precisely which parameters are applicable to define the benefit for the patients.

The data section lacks statements regarding the parameters supposed to be analyzed in the results:

Were there defined resection standards in the centers? How was the tumor diameter determined (line #127); sonography, CT, MRT with a fixed specimen? How was the blood loss measured? Based on which imaging was the vessel contact determined? Which kind of chemotherapy was administered prior to surgery? Were there patients with immunotherapy? What is the definition of “prolonged pain” (table #1)?

Is there really a gain in information if stay on ICU and hospital stay are analyzed without defined criteria for patient stay on ICU/in hospital? One may see a decision bias in favor of an elderly patient.

Wouldn’t it be better to apply Median and Range instead of Mean and SD in such an inhomogeneous sample with such a small sample size.

It remains unclear to the reader what may be the meaning of table #1, given that according to the title of the manuscript younger patients are compared versus elderly patients.

Also unclear is the gain of information from figures 1 and 2 if one finds the same values as numbers already in the tables (table #1 as well as summative in table #2)?

Discussion: While the introduction is very focused, the authors get somehow lost in their discussion. The history of the guidelines is for sure not in the focus of the manuscript (line #176-224).

The scientific gain of the surely interesting study compared to already existing evaluations needs to be presented more distinctive.

Unfortunately, the reader is confronted with a sloppy editing, repeated citations inconsistent in their presentation (line #204) versus line #274 for example, also BMI with kg/cm2 and not kg/m², various typos like Chemoterapy (table 1), blood trasfusion etc.

In the here presented form a publication in a high ranked journal like Cancers cannot be supported.

Comments on the Quality of English Language

final editing must be improved

Author Response

Rev 1

Robotic major hepatectomy is a very relevant topic, given current results promise significant advantages for the patients.

  1. The introduction is held pleasantly short and targets the topic. The authors should be more precise in the wording of the target of their study (line #62/63), describing more precisely which parameters are applicable to define the benefit for the patients.

Answer: we modified the following sentences

 “The benefits of robotic liver resection are emerging thanks to studies that confirm the role of MILS in reducing post-operative complications, better post-operative recovery and shorter hospitalization. These represent a real advantage especially in elderly pa-tients who can benefit from a quick return home. To date, the literature is poor in stud-ies that have investigated the role of robotic major hepatectomy in elderly patients. The aim of this study was to analyze the feasibility and safety of robotic major liver resection in elderly patients, considering intraoperative and post-operative short term out-comes.”

  1. The data section lacks statements regarding the parameters supposed to be analyzed in the results:

Answer: “Preoperative, intraoperative, and postoperative data were retrospectively analyzed and compared in both groups”.

Following, each paragraph defines the parameters analyzed in the results.

  1. Were there defined resection standards in the centers? How was the tumor diameter determined (line #127); sonography, CT, MRT with a fixed specimen? How was the blood loss measured? Based on which imaging was the vessel contact determined? Which kind of chemotherapy was administered prior to surgery? Were there patients with immunotherapy? What is the definition of “prolonged pain” (table #1)?

Answer: we modified the following sentences

“Liver segmentation anatomy and liver resection type were defined using the Couinaud classification and the Brisbane 2000 terminology, respectively. Major liver resection was considered a resection of 3 or more contiguous Couinaud segments.”

“CT scan allowed to define tumor diameter and vascular contact.”

“Preoperative chemotherapy was based on tumor biology”.

“Prolonged pain was defined as continued use of analgesics beyond the third postoperative day”.

“Blood loss was measured as the difference between the fluid present in the suction receptor and the lavage solution infused into the abdominal cavity”

  1. Is there really a gain in information if stay on ICU and hospital stay are analyzed without defined criteria for patient stay on ICU/in hospital? One may see a decision bias in favor of an elderly patient.

Answer: the criteria depend on the experience and structural organization of each centre. Hospital stay was similar between two group and also ICU stay, after statistic adjusting

  1. Wouldn’t it be better to apply Median and Range instead of Mean and SD in such an inhomogeneous sample with such a small sample size.

Answer: the statistician chose to use Mean and SD to better represent the results.

  1. It remains unclear to the reader what may be the meaning of table #1, given that according to the title of the manuscript younger patients are compared versus elderly patients.

Answer: Data were collected from a multicentric retrospective database including 1070 consecutive robotic liver resection for benign and malignant liver tumors. From the entire series, 131 were robotic major hepatectomy. We think it is interesting to show the characteristics of all major liver resections before comparing younger and older patients.

  1. Also unclear is the gain of information from figures 1 and 2 if one finds the same values as numbers already in the tables (table #1 as well as summative in table #2)?

Answer: We agree. We have eliminated the data from the table already shown in the figure.

  1. Discussion: While the introduction is very focused, the authors get somehow lost in their discussion. The history of the guidelines is for sure not in the focus of the manuscript (line #176-224).

Answer: we deleted the paragraph.

  1. The scientific gain of the surely interesting study compared to already existing evaluations needs to be presented more distinctive.

Answer: we modified the following sentences

“The aim of this study was to analyze the feasibility and safety of robotic major liver resection. These benefits were confirmed also for elderly patient after a comparative analysis with the young population”.

  1. Unfortunately, the reader is confronted with a sloppy editing, repeated citations inconsistent in their presentation (line #204) versus line #274 for example, also BMI with kg/cm2 and not kg/m², various typos like Chemoterapy (table 1), blood trasfusion etc.

Answer: we made the changes.

Reviewer 2 Report

Comments and Suggestions for Authors

Comments to the authors:

The aimed of the study was to analyze the feasibility and safety of robotic major liver resection in elderly patients. We congrats the authors, however some concerns need to be addressed before consideration for publication.

Major comments

1.     One of the major concerns is that although the mean age of the entire cohort was 64 years, the cutoff for elderly patients was considered to be 65. With this in mind, could you please elaborate on why you considered ≥65 years as the criterion for elderly patients? Also consider adding this in methods section.

2.     Simple Summary section: Why is this sentence relevant in your manuscript? “Despite these advantages, up to now in literature no evidence of benefits of the robotic approach in liver surgery has been clearly defined”. It would make sense if the aim of the study was to investigate regarding benefits of robotic vs laparoscopic or open approaches. If not, and you affirm that there is no evidence of benefits for robotic approach, why would you go one step ahead and try to find out if an approach with not proven benefit is feasible in elderly patients? I would suggest removing or rephrasing that sentence.

3.     Introduction section. Even though the study is interesting and valid, it is not well stablished the need to investigate the feasibility and safety of robotic major liver resection in elderly patients, nor whether that particular topic was explored in previous literature. I would recommend adding this in the introduction.

4.     Introduction section. The following phrase is not clear: “The aim of our study is to confirm this kind of benefits in major liver resection also in elderly patients.” As the previous statement verses about MILS, it is not clear if the quoted phrase is referring to MILS or robotic surgery. I would suggest having a similar wording (more specific) as the aim stated in the simple summary and abstract: “the aim of this study was to analyze the feasibility and safety of robotic major liver resection in elderly patients.”

5.     Results section. You mentioned that you matched the 2 groups for ASA and Charlson comorbidity score. Could you please explain which statistical test or analysis was used to do so? If applicable, include Confidence Interval or OR. Would suggest including this in methods section and table 4.

6.     A discussion is an adequate interpretation of the results, however, the first paragraph of discussion section (lines 176 – 225) describes historical milestones about MILS development. Please consider deleting that paragraph.

7.     Discussion section, line 226, the phrase “Our study shows the benefits of robotic approach in major liver resection also in elderly patients.” is overreaching as the aim of the study was to demonstrate the feasibility and safety of robotic major liver resection (RMLR) in elderly patients. Please consider rephrasing that sentence.

8.     The main purpose of this study was to demonstrate the feasibility and safety of RMLR in elderly patients, for which the elderly population was compared to the younger one; however, the second paragraph (lines 226 – 262) focuses on describing the results in the entire cohort (elderly + young) and compares those results to the ones from other studies that only recruited elderly individuals (e.g. Yoshino et al. Surg Endosc. 2023; 37:6228) or the results of the elderly cohort in another studies. (e.g., Sucandy et al. Am. Surg. 2021; 87: 114). Please when comparing to those studies, use the results obtained from the elderly population in your manuscript.

9.     There was a significant difference in the operative time between the young and the elderly population, with mean operative times of 312 min vs 358 min, respectively (p= 0.050, deemed significant according to the manuscript: “A p value of 0.05 or less was considered statistically significant”.); however, this finding was not mentioned in the results section and not explained in the Discussion section. Could you please elaborate on why was this not included (maybe considered not relevant) or elaborate more about it in the manuscript?

Minor comments

1.     The format of tables and figures should be improved as publication quality, i.e.:

a.     Please provide a complete Abbreviation section under each table and figure.

b.     Do not use bold type apart from the first row. However, if you are using bold type for emphasis, explain the reason below that table.

c.     Each table should fit in one page.

d.     Consider using “italics” for statistically significant P-values (<0.050) for emphasis and easier understanding.

2.     Please place a complete and adequate Abbreviation section at the beginning of the manuscript after Abstract.

3.     Simple Summary, page 1, line 30, replace the word “is” by “was.”

4.     Abstract section, page 1, line 33, replace the word “is” by “was.”

5.     Introduction section, page 2, line 60, replace the word “papers” by “studies”.

6.     Discussion section, page 9, line 252, replace the word “or” by “our”.

Comments on the Quality of English Language

Good

Author Response

Rev 2

Comments to the authors:

The aimed of the study was to analyze the feasibility and safety of robotic major liver resection in elderly patients. We congrats the authors, however some concerns need to be addressed before consideration for publication.

Major comments

  1. One of the major concerns is that although the mean age of the entire cohort was 64 years, the cutoff for elderly patients was considered to be 65. With this in mind, could you please elaborate on why you considered ≥65 years as the criterion for elderly patients? Also consider adding this in methods section.

Answer: We used a standard definition for elderly patient >65.   

We have added to method: “The definition of elderly people concerns patients over 65 years of age”.

  1. Simple Summary section: Why is this sentence relevant in your manuscript? “Despite these advantages, up to now in literature no evidence of benefits of the robotic approach in liver surgery has been clearly defined”. It would make sense if the aim of the study was to investigate regarding benefits of robotic vs laparoscopic or open approaches. If not, and you affirm that there is no evidence of benefits for robotic approach, why would you go one step ahead and try to find out if an approach with not proven benefit is feasible in elderly patients? I would suggest removing or rephrasing that sentence.

Answer: “The benefits of robotic liver resection are emerging thanks to studies that confirm the role of MILS in reducing post-operative complications, better post-operative recovery and shorter hospitalization”.

  1. Introduction section. Even though the study is interesting and valid, it is not well stablished the need to investigate the feasibility and safety of robotic major liver resection in elderly patients, nor whether that particular topic was explored in previous literature. I would recommend adding this in the introduction.

Answer: we modified the following sentences

“These represent a real advantage especially in elderly patients who can benefit from a quick return home. To date, the literature is poor in studies that have investigated the role of robotic major hepatectomy in elderly patients”.

  1. Introduction section. The following phrase is not clear: “The aim of our study is to confirm this kind of benefits in major liver resection also in elderly patients.” As the previous statement verses about MILS, it is not clear if the quoted phrase is referring to MILS or robotic surgery. I would suggest having a similar wording (more specific) as the aim stated in the simple summary and abstract: “the aim of this study was to analyze the feasibility and safety of robotic major liver resection in elderly patients.”

Answer: we modified the following sentences

“The aim of this study was to analyze the feasibility and safety of robotic major liver resec-tion in elderly patients, considering intraoperative and post-operative short term out-comes”.

  1. Results section. You mentioned that you matched the 2 groups for ASA and Charlson comorbidity score. Could you please explain which statistical test or analysis was used to do so? If applicable, include Confidence Interval or OR. Would suggest including this in methods section and table 4.

Answer: Statistical analysis section “Multivariate logistic and linear regression were used to evaluate the independent associa-tion between age category and dichotomous (logistic) or continuous (linear) variables ad-justing for ASA and Charlson comorbidity score that were significantly different between patients younger and older than 65 years”.

  1. A discussion is an adequate interpretation of the results, however, the first paragraph of discussion section (lines 176 – 225) describes historical milestones about MILS development. Please consider deleting that paragraph.

Answer: we deleted the paragraph

  1. Discussion section, line 226, the phrase “Our study shows the benefits of robotic approach in major liver resection also in elderly patients.” is overreaching as the aim of the study was to demonstrate the feasibility and safety of robotic major liver resection (RMLR) in elderly patients. Please consider rephrasing that sentence.

Answer: we modified the following sentences

“The aim of this study was to analyze the feasibility and safety of robotic major liver resection. These benefits were confirmed also for elderly patient after a comparative analysis with the young population”.

  1. The main purpose of this study was to demonstrate the feasibility and safety of RMLR in elderly patients, for which the elderly population was compared to the younger one; however, the second paragraph (lines 226 – 262) focuses on describing the results in the entire cohort (elderly + young) and compares those results to the ones from other studies that only recruited elderly individuals (e.g. Yoshino et al. Surg Endosc. 2023; 37:6228) or the results of the elderly cohort in another studies. (e.g., Sucandy et al. Am. Surg. 2021; 87: 114). Please when comparing to those studies, use the results obtained from the elderly population in your manuscript.

Answer: we made correction, comparing our elderly population with elderly cohort in literature.

  1. There was a significant difference in the operative time between the young and the elderly population, with mean operative times of 312 min vs 358 min, respectively (p= 0.050, deemed significant according to the manuscript: “A p value of 0.05 or less was considered statistically significant”.); however, this finding was not mentioned in the results section and not explained in the Discussion section. Could you please elaborate on why was this not included (maybe considered not relevant) or elaborate more about it in the manuscript?

Answer: Generally, p value < 0.05 was considered statistically significant. A p value = 0.05 is the cut off for significance. For this reason we don’t included this in discussion.

Reviewer 3 Report

Comments and Suggestions for Authors

In this manuscript, the authors retrospectively reviewed 131 patients  (<65 years old and ≥65 years old) who were underwent robotic liver resection resections to analyze the feasibility and safety of major liver resection in elderly patients.  Postoperative data were similar between two groups. So, the authors concluded that  Robotic major liver resection in elderly patients is feasible with satisfy short-term outcomes. Some concerns about that:

1. The new findings or information (if any) over the pubished data and Consensus Guidelines shoud be emphasized (such as indictions, patients' selection, or experiens in peroperative managements, etc.). It seems just to retrospectively reviewed the postoperative data of 131 patients, and to conclude that "it is okay".

2. 131 patients  (<65 years old and ≥65 years old) were underwent Robotic major liver resections for various indications form 9 European Hospital Centers. How to avoid the bias in selecting patients? and the reason that using 65 ys old as the cut-off value to define "elderly patient" shoule be present.

3.  37 (28.2%) patients with HCC were erolled, most of them might have chronic liver disease or liver cirrhosis background which would influnce the safty, blood loss and operating time of robotic  liver resetion, as well as the liver function recovery. The etiology and severity of liver disease, and the selection criteria should be discussed.

Comments on the Quality of English Language

English writing needs to be further improved.

Author Response

Rev 3

In this manuscript, the authors retrospectively reviewed 131 patients  (<65 years old and ≥65 years old) who were underwent robotic liver resection resections to analyze the feasibility and safety of major liver resection in elderly patients.  Postoperative data were similar between two groups. So, the authors concluded that  Robotic major liver resection in elderly patients is feasible with satisfy short-term outcomes. Some concerns about that:

  1. The new findings or information (if any) over the pubished data and Consensus Guidelines shoud be emphasized (such as indictions, patients' selection, or experiens in peroperative managements, etc.). It seems just to retrospectively reviewed the postoperative data of 131 patients, and to conclude that "it is okay".

Answer:  we modified the following sentences

“Our study showed the feasibility and safety of robotic major liver resection. These benefits were confirmed also for elderly patient after a comparative analysis with the young population.”

Indications, patients' selection, experience in perioperative managements were showed in methods.

  1. 131 patients  (<65 years old and ≥65 years old) were underwent Robotic major liver resections for various indications form 9 European Hospital Centers. How to avoid the bias in selecting patients? and the reason that using 65 ys old as the cut-off value to define "elderly patient" shoule be present.

Answer: we modified the following sentences

“The definition of elderly people concerns patients over 65 years of age”.

The enrollment of all liver diseases, both benign and malignant, represents a limitation of this study. In this way we enrolled 131 patients of which 56 were elderly. Due to the limited number we wanted to include all liver diseases.

  1. 37 (28.2%) patients with HCC were erolled, most of them might have chronic liver disease or liver cirrhosis background which would influnce the safty, blood loss and operating time of robotic  liver resetion, as well as the liver function recovery. The etiology and severity of liver disease, and the selection criteria should be discussed.

Answer: we modified the following sentences

“Many patients, 28.3%, had underlying liver fibrosis with a mean MELD score of 6.8 with satisfactory liver function. Some authors have demonstrated that RLR can be used for major liver resection in cirrhotic disease obtaining less postoperative pain and shorter hospital stays without changing oncological outcomes”.

Reviewer 4 Report

Comments and Suggestions for Authors

Congratulation for the authors in writing this paper in an interesting area of liver resection " Robotic Major Hepatectomy in elderly patient ". The quality of paper is goodthe study design is appropriate to anser the research and the metods are sufficiently described to allow the study to be repeated . The presentation of the work is clear.The reference list cover the relevant literature adequately.

My recommendation : accept